# Diagnosis and Management of Type 1 Sialidosis: Clinical Insights from Long-Term Care of Four Unrelated Patients

**DOI:** 10.3390/brainsci10080506

**Published:** 2020-08-01

**Authors:** Antonietta Coppola, Marta Ianniciello, Ebru N. Vanli-Yavuz, Settimio Rossi, Francesca Simonelli, Barbara Castellotti, Marcello Esposito, Stefano Tozza, Serena Troisi, Marta Bellofatto, Lorenzo Ugga, Salvatore Striano, Alessandra D’Amico, Betul Baykan, Pasquale Striano, Leonilda Bilo

**Affiliations:** 1Department of Neuroscience, Reproductive and Odontostomatological Sciences, Federico II University, 80131 Naples, Italy; marta.ianniciello@gmail.com (M.I.); marcelloesposito@live.it (M.E.); ste.tozza@gmail.com (S.T.); serena.troisi@gmail.com (S.T.); marta.bellofatto89@gmail.com (M.B.); sstriano1@gmail.com (S.S.); leda.bilo@gmail.com (L.B.); 2Department of Neurology, Clinical Neurophysiology Unit, Faculty of Medicine, Istanbul University, 34116 Istanbul, Turkey; ebruvanli@gmail.com (E.N.V.-Y.); betul.baykan.baykal@gmail.com (B.B.); 3Department of Neurology, Koc University Hospital, 34010 Istanbul, Turkey; 4Eye Clinic, Multidisciplinary Department of Medical, Surgical and Dental Sciences, University of Campania L. Vanvitelli, 80131 Naples, Italy; settimio.rossi@unicampania.it (S.R.); francesca.simonelli@unicampania.it (F.S.); 5Unit of Genetic of Neurodegenerative and Metabolic diseases, Fondazione IRCCS Istituto Neurologico Carlo Besta, 20133 Milano, Italy; Barbara.Castellotti@istituto-besta.it; 6Department of Advanced Biomedical Sciences, Federico II University, 80131 Naples, Italy; lorenzo.ugga@gmail.com (L.U.); damicoalex@tiscali.it (A.D.); 7Department of Neurosciences, Rehabilitation, Ophthalmology, Genetics, Maternal and Child Health (DiNOGMI), University of Genoa, 16147 Genoa, Italy; strianop@gmail.com; 8Pediatric Neurology Unit, IRCCS Istituto Giannina Gaslini, 16147 Genoa, Italy

**Keywords:** type-1-sialidosis, myoclonus, brain MRI, cherry-red spot, giant SEP, jerk-locked back averaging analysis

## Abstract

*Background*: Sialidosis is a rare autosomal recessive disease caused by *NEU1* mutations, leading to neuraminidase deficiency and accumulation of sialic acid-containing oligosaccharides and glycopeptides into the tissues. Sialidosis is divided into two clinical entities, depending on residual enzyme activity, and can be distinguished according to age of onset, clinical features, and progression. Type 1 sialidosis is the milder, late-onset form, also known as non-dysmorphic sialidosis. It is commonly characterized by progressive myoclonus, ataxia, and a macular cherry-red spot. As a rare condition, the diagnosis is often only made after few years from onset, and the clinical management might prove difficult. Furthermore, the information in the literature on the long-term course is scarce. *Case presentations*: We describe a comprehensive clinical, neuroradiological, ophthalmological, and electrophysiological history of four unrelated patients affected by type 1 sialidosis. The long-term care and novel clinical and neuroradiological insights are discussed. *Discussion and conclusions*: We report the longest follow-up (up to 30 years) ever described in patients with type 1 sialidosis. During the course, we observed a high degree of motor and speech disability with preserved cognitive functions. Among the newest antiseizure medication, perampanel (PER) was proven to be effective in controlling myoclonus and tonic–clonic seizures, confirming it is a valid therapeutic option for these patients. Brain magnetic resonance imaging (MRI) disclosed new findings, including bilateral gliosis of cerebellar folia and of the occipital white matter. In addition, a newly reported variant (c.914G > A) is described.

## 1. Background 

Sialidosis is a rare autosomal recessive lysosomal storage disease (approximate prevalence of 1/5,000,000-1/1,500,000 live births) caused by a deficiency of neuraminidase (sialidase) due to mutations in the *NEU1* gene located on chromosome 6p21.3 [1]. Patients have an impaired degradation of glycoproteins and subsequent accumulation of sialic acid-containing oligosaccharides and glycopeptides into the tissues. 

Sialidosis is divided into two clinical entities that can be distinguished according to the age of onset, clinical features and progression. Type 1 sialidosis is a milder, late-onset form characterized by visual defects, macular cherry-red spot, myoclonus, ataxia, and seizures. Patients with type 2 sialidosis show a more severe, early-onset disease with predominant visceral features associated with multiplex dysostosis, intellectual disabilities, and hepatosplenomegaly. 

The characteristic cherry-red spot due to accumulation of metabolic substrates in the macular area represents a cardinal sign shared by both forms and has long been considered as the most typical marker of sialidosis. However, patients presenting without the macular cherry-red spot have been reported and clinical presentation might be subtle at the onset [2,3]. As myoclonus is the key symptom being present in 100% of the patients, type 1 sialidosis is considered among the progressive myoclonus epilepsies (PMEs). This group of conditions are neurophysiologically characterized by cortical hyperexcitability demonstrated by the evidence of a giant somatosensory evoked potential (SEP); the presence of the long loop reflex (LLR), also known as C-reflex at rest; and a time-locked cortical potential preceding electromyography (EMG) bursts evidenced by the jerk-locked back averaging analysis (JL-Back AVG) [4,5].

Pharmacological treatment is symptomatic as for the other PMEs and typically requires a polytherapy of anti-seizure medications (ASMs) [5]. However, the landscape of available drugs is considerably limited because some ASMs are contraindicated or even deleterious in these conditions [6]. 

Long-term comprehensive follow-up descriptions are missing for these patients. We report the clinical, neuroradiological, ophthalmological, and electrophysiological long course of four unrelated patients affected by type 1 sialidosis and bearing compound heterozygosity of *NEU1.* We will discuss the peculiarities encountered over a long follow-up (up to 30 years).

## 2. Case Presentations

### 2.1. Patient #1

A 43-year-old Italian female, born from healthy unrelated parents. Her family history showed two granduncles with epilepsy; one died due to status epilepticus. The patient was healthy until the age of 13 years when she started complaining of difficult reading because of oscillopsia. Soon after, frequent falls due to gait instability occurred. At age 15 years, she was referred for her first tonic-clonic seizure (TCS). The neurological examination showed mild gait ataxia, horizontal nystagmus, mild dysarthria, bilateral dysmetria at finger-nose tapping, fine myoclonus involving the upper arms at rest and with action, generalized hypotonia, and hyperactive deep tendon reflexes. Her intelligence quotient (IQ) was within the normal range. At age 17 years, ataxia and myoclonus worsened and involved all the limbs, resulting in difficulty standing up and walking independently. Dysarthria also became severe. Her eye movements showed a flutter in the primary position of the sight. At age 21 years, the woman was wheelchair-bound; the myoclonus affecting all four limbs and perioral muscles impeded even the simple motor tasks. The speech was unintelligible because of dysarthria and palatal and tongue myoclonus. Her sight became poor and opsoclonus appeared. The IQ was confirmed within the normal range. At the age of 22, during a hormonal treatment because of amenorrhea, she experienced severe worsening of myoclonus that was triggered by even mild tactile, acoustic, and visual stimuli leading to a myoclonic status epilepticus, resistant to clonazepam. The status ceased after she was put in a dark stimuli-free room. At the same age, TCSs were rare (yearly frequency) while massive myoclonus was frequent, mostly related to the menstrual cycle and very invalidating. At the age of 24, she was bedridden because of severe asthenia and distal atrophy; the dysarthria and sight had significantly worsened. In the following years, several complications arose, including obstructive apnea syndrome, hypothyroidism, kidney and biliary stones, bilateral cataracts, and glaucoma. At her last follow up at the age of 42 years, the patient was cognitively intact. She was bedridden with a spastic tetraparesis and blind. 

*Treatment*: Over the years, the patient did try several medications, including phenobarbital (PB) valproate (VPA), clonazepam (CLZ), clobazam (CLB), levetiracetam (LEV), topiramate (TPM), and zonisamide (ZNS) in various combinations and doses, reporting only mild and transient benefits. During the last two years, perampanel (PER; 8 mg/day) added on to the therapeutic regimen (PB 100 mg/day, VPA 600 mg/day, CLZ 4 mg/day, LEV 3000 mg/day) resulting in reduced convulsive seizure frequency and improved myoclonus (myoclonus seizures dropped to isolated short-lasting 5–10 episodes per day, while previously they were up to 40–50 per day, often in clusters). This allowed an improvement in the quality of her life, especially with regards to eating and speaking.

*Neurophysiological Investigations:* At the onset, the electroencephalography (EEG) did not show epileptiform discharges. The JL-Back AVG detected a biphasic (positive–negative) potential over the central electrodes, which resulted in time-locked cortical potential with the myoclonus registered from the right leg. SEP showed a giant potential (P25-N30: 18.8 μV) [7]; the LLR was also present. Afterwards, the EEG background activity slowed down significantly. At the age of 22, the EEG showed frequent high amplitude and diffuse spike and slow-wave discharges not associated with clinical manifestations (Appendix A). At her last follow up (42 years old), the EEG showed a dominant theta (6–7 Hz) background activity, and high voltage delta discharges intermixed with epileptiform abnormalities (spike and polyspike wave discharges over the temporal areas) (Appendix A).

*Ophthalmological Examinations*: At the age of 13 years, fundus oculi was unremarkable and no cherry-red spot was observed even in consecutive examinations up to the age of 20 years. From that time, opsoclonus (Appendix A) and cataracts proved difficult to ascertain its presence. At 24 years old, after a significant visual drop, visual evoked potentials (VEP) were performed, demonstrating bilateral severely impaired potentials (very low amplitude and longer latency). A severe cataract was also present in both eyes, impeding the fundus oculi examination.

*Brain Images*: The first brain MRI performed at the age of 14 years was normal. After 10 years, at 24 years, the brain MRI showed cortical atrophy and a mild reduction of the inferior and middle cerebellar peduncles. The last brain MRI (40 years) showed frontal hyperostosis, bilateral hyperintensities of the peritrigonal, and the deep occipital white matter. The optic nerves and the chiasm were thinned. A severe and diffuse cerebellar atrophy associated with hyperintensity of the folia of part of the hemispheres and the vermis (declive, folium, tuber) were also evident (Figure 1).

Cerebral 18F-fluorodeoxyglucose (FDG)-positron emission tomography (PET), performed at the age of 29 years, showed reduced metabolism over the occipital cortex, bilaterally.

*Additional Investigations:* Muscle biopsy performed at the age of 30 years showed a dystrophic pattern and an altered glycogen distribution. However, the immunohistochemistry showed a normal pattern for dystrophin, αβγδ sarcoglycan, dysferlin, spectrin, caveolin, and merosin. The measurement of the cytochrome C oxidase activity resulted as being slightly reduced. Alpha neuraminidase activity was severely reduced: 0.55 nM/h/mg (normal values: 100–390 nM/h/mg); urinary bound sialic acid was 93 nM/creatine M; normal value: M <60 nM/creatine M). An abdominal echography (aged 40 years) showed liver and spleen hyperplasia. 

*Genetics*: The patient underwent different molecular investigations that excluded Lafora disease (laforin—*EPM2A* and malin—*NHLRC1*), Unverricht–Lundborg disease (cystatin B, *CSTB)*, myoclonic epilepsy with ragged red fibers (MERFF), mitochondrial encephalopathy, lactic acidosis, and stroke-like episodes (MELAS), ceroid lipofuscinoses, hereditary ataxias, opsoclonus–myoclonus paraneoplastic syndrome, Kinsbourne disease, and Ramsay–Hunt disease. At the age of 36 years, direct sequencing of coding regions of *NEU1* gene showed compound heterozygosity with a missense mutation c.982G > A (p.Gly328Ser rs534846786) and a deletion c.1208delG (p.Ser403ThrfsTer85 rs1301852124), reported in Mouna et al. (PME23-1) [8].

### 2.2. Patient #2

A 28-year-old Italian female born from healthy consanguineous (first degree cousins) parents. Her personal history was uneventful until the age of 13 when frequent falls due to gait instability were reported. At that time, her parents noted a fine tremor affecting the upper limbs and a few months later also her legs. She also presented with hemeralopia. At the age of 17 years, the neurological examination showed ataxic gait, dysmetria of rapid eye movements and finger–nose tapping, and nystagmus in the horizontal sight towards the right. Mild postural and action myoclonus were visible in all four limbs. Three years later, aged 20 years, there was a worsening of the myoclonus, appearance of ataxia, and hyposthenia of all limbs, which caused difficulty in walking. At the age of 23, the action myoclonus severely affected her daily life activities and she lost her ability to walk independently. The visual acuity also worsened. Cranial nerves showed bilateral mydriasis, hypometria of the eye movements, and horizontal nystagmus. Dysmetria and dysdiadochokinesia were also present. Dysarthria severely affected the intelligibility of her language. At the age of 24 years, the patient got married and at the age of 25 got pregnant but spontaneously aborted the fetus at the 13th week of gestation. After 4 months, she had her first TCS followed by a second one after 1 month. At that time, multiple episodes of massive myoclonus (starting from the shoulder and spreading to the arms and legs) with preserved consciousness occurred many times a day, sometimes triggered by eye closure and by emotional stress. She is now 28 years old, wheelchair-bound, and able to walk if assisted but only for a short distance; she is severely dysarthric and her sight has significantly worsened. 

*Treatments:* At the onset, although she never suffered from clinical seizures, she started treatment with VPA first and carbamazepine (CBZ) was added on later. Subsequently, she was treated with LEV, piracetam (PIR), acetazolamide, and ZNS, which were used in combination. At the age of 26 years, after a significant worsening of her myoclonus (multiple/day sudden, brief and massive myoclonic attacks), PER 4 mg/day was added onto the treatment (CLZ 3 mg/day, acetazolamide 250 mg/day, and LEV 1500 mg/day) with a complete disappearance of the myoclonic seizures. Unified myoclonus rating scale (UMRS) was performed to evaluate the myoclonus severity, giving a total score of 144 before PER treatment. After 3 months, a significant improvement was reported by the patient and was also detected at the clinical evaluation with an improved UMRS score of 100. The benefit was particularly evident for daily life activities such as eating and writing.

*Neurophysiological Investigations:* At onset (13 years old), the EEG showed a normal background activity with interictal generalized epileptiform discharges. At the age of 27 years, the EEG showed a background activity characterized by bilateral irregular alpha rhythm, as well as drug-related rapid rhythms; frequent discharges characterized by diffuse polyspikes accompanied by intense myoclonus affecting all four limbs and face were recorded spontaneously or provoked by intermittent light stimulations and eye closure (Figure 2). After the introduction of PER, these discharges completely disappeared and so did the myoclonic seizures (Figure 2).

VEP findings were consistent with a dysfunction of the visual pathway (Figure 3). Motor-evoked potential (MEP) also disclosed a dysfunction of the cortico-spinal tract of the lower arms. SEP revealed a giant potential (Figure 3) [7]. Superficial EMG did not disclose spontaneous muscle activity.

*Ophthalmological Examinations*: The first examination was performed at 15 years and disclosed normal functioning. A second ophthalmologic evaluation was performed at the age of 21. The normal size of the optic nerves and normal pupillary reflexes were noted; visual acuity was 20/50 for both eyes. Fundus oculi disclosed a cherry-red spot in the right eye and retinal pigment epithelium dystrophy in the left eye. Farnsworth test showed difficulties for blue-yellow axis and red-green axis in the right eye, and in red-green axis in the left eye.

After a visual drop, at the age of 27 years, the patient underwent a complete ophthalmological examination including best-corrected visual acuity (BCVA) measured with an Early Treatment Diabetic Retinopathy Study (ETDRS) chart at 2 meters; slit-lamp biomicroscopy examination, ocular fundus exam with retinography, and finally an optical coherence tomography (OCT) with Heidelberg Engineering were performed. She had the following BCVA: 45 letters (20/63) in the right eye and 33 letters (20/100) in the left eye. Slit lamp examination showed, in both eyes, a slight diffuse opacity of the corneal endothelium and incipient lens cortical opacities. There was no nystagmus. Examination of the ocular fundus showed, bilaterally, a macular cherry-red spot with diffuse dystrophy of the retinal pigment epithelium in the middle periphery and a normal optic disc (Figure 4). Spectral domain OCT showed a thickening with increased reflectivity of ganglion cell layer (GCL) in both eyes (Figure 4).

*Brain Images*: The first brain MRI, performed at age 17 years, showed enlarged pericerebellar cerebrospinal fluid (CSF) spaces, mostly around the cerebellar hemispheres. Cerebella H^1^ spectroscopy was executed and an *N*-acetylaspartate (NAA) signal resulted slightly below the normal range as per initial cortico-subcortical cerebellar atrophy. Six years later, aged 23, the cerebellar atrophy and CSF spaces and fourth ventricle dilation were confirmed (see Figure 1).

*Additional Investigations:* At age 17 years, the neuraminidase enzymatic activity was absent in the fibroblasts. Neuropsychological examination performed at the same age revealed a mild isolated short span verbal memory deficiency.

*Genetics:* The patient underwent molecular analysis for Unverricht–Lundborg disease (cystatin B, *CSTB)* and *MERFF* (c.8344A>G and c.8356 T>C variants), and PME1b (*PRICKLE1)* that did not disclose pathogenic variants. At the age of 18 years, the direct sequencing analysis of coding regions of *NEU1* disclosed a heterozygous compound for the described missense mutations c.272T>G (p.Leu91Arg rs104893972) and c.982G>A (p.Gly328Ser- rs534846786). One of her brothers carried the heterozygous missense mutation c.272T>G (p.Leu91Arg); another carried the heterozygous missense mutation c.982G>A (p.Gly328Ser), and the third one carried none of them. 

### 2.3. Patient #3

A 41-year-old Turkish female born to consanguineous parents (third-degree relatives); her 6-year-old cousin suffered from epilepsy but no further details were available. At 17 years of age, she presented with complaints of bilateral whole-body tremor, myoclonic jerks, and imbalance, which was prominent at the left hand, especially in the mornings. TCSs began in the same year, and tremor also worsened. Her convulsions were triggered by excitation. Despite treatment with VPA, CLZ, primidone, and PIR, the seizures recurred at least once a month. Speech disorder and forgetfulness appeared after 3-4 years. At the age of 20 years, her neurological examination revealed dysarthric speech, ataxia, and positive and negative myoclonus affecting all four limbs. Myoclonus was generalized, synchronized, and worsened with action. She stood up with bilateral support; myoclonus increased when she stepped forward; and after a few steps, she tended to fall due to negative myoclonus. Her mental status was within normal limits and she succeeded in university entrance exams. The neuropsychological evaluation revealed impairments in executive functioning (frontal signs) such as increase in interference time, deficiency in category formation and changing, perseverations, and decrease in mental flexibility. At last follow up (38 years old), the patient was confined to wheelchairs due to severe ataxia while her TCSs had ceased. 

*Treatments:* TCSs ceased in the last year after ZNS 400 mg/day was added to her treatment, which previously consisted of VPA 1500 mg/day, primidone 500 mg/day, LEV 3000 mg/day, and CLZ 2 mg/day. 

*Neurophysiological Examinations:* EEG evaluation revealed mild and a general slowing of the background activity prominent in the anterior hemispheres and hypersynchronization at the vertex. JL-Back AVG analysis disclosed a biphasic right parietal wave approximately 20 ms before myoclonus potentials recorded from superficial electrodes located on the left forearm extensor muscles; median SEP evaluation revealed bilateral giant cortical responses (Figure 3) [7]. With right median nerve stimulation, a moderate latency LLR was recorded from right thenar muscles. With double stimulus transcranial magnetic stimulation technique, no inhibitory response (which is expected under normal conditions) was seen after the second stimulus at 100 ms interstimulus interval. Nerve conduction studies, VEP, and tibial SEP evaluations were normal. 

*Ophthalmological Examinations:* Repeated ophthalmological evaluation during the course confirmed absence of the typical macular “cherry-red spot”.

*Brain Images*: At 17 years old, the brain MRIs were normal, but the last one performed at the age of 40 years showed cerebral atrophy, mostly involving the parietal and occipital lobes and cerebellar atrophy. The globi pallidi appeared hypointense. The enlargement of cerebellar subarachnoid spaces indicated the presence of atrophy of vermis and both hemispheres (see Figure 1).

*Additional Investigations:* Skin biopsy, which was evaluated twice; muscle biopsy; duodenal biopsy; detailed biochemical analyses; and lactate value were normal. Antigliadin antibodies were negative. 

*Genetics*: In the first years, she underwent different genetic testing (Sanger sequencing) in order to exclude Unverricht–Lundborg disease (cystatin B, *CSTB),* PME with renal failure *(SCARB2),* Lafora disease (laforin—*EPM2A*, and malin—*NHLRC1*). All of them resulted normal. The genetic analysis of the *NEU1* gene (30 years old) identified a described heterozygous missense mutation c.914G>A (p.Arg305His rs774362886) and a novel heterozygous deletion of a nucleotide c.625delG, which determines a frameshift mutation p.Glu209SerfsTer94. The segregation analysis was only performed in the mother, who showed the novel frameshift variant c.625delG variant. The genetic details of this patient have already been reported in Muona et al. (PME 87-1) [8].

### 2.4. Patient #4

An 18-year-old Italian female born from healthy unrelated parents. Family history was positive for epilepsy (an uncle from the maternal side suffered from epilepsy). Her personal history was uneventful until the age of 12, when her parents noticed a fine postural tremor affecting the upper limbs. From the age of 16 years, she also reported imbalance while walking, with myoclonic jerks prominent at the left foot and left hand. At the age of 17 years, the patient came under our care. Her neurological examination showed mild postural and action myoclonias in all four limbs, prominent on the left side. She was able to walk only with support on one side; however, myoclonus increased while walking. At present, the patient is 18 years old, and she can walk unassisted only for a short distance. The myoclonus is particularly severe during her menses. 

*Treatments:* The patient started treatment with LEV up to 1 g per day with a short-term benefit. Then, CLZ 1 mg/day was added at first, and acetazolamide 250 mg/day was added later. In the last 6 months of follow up, PER up to 4 mg was added, which showed a significant benefit, particularly with allowing her to walk again independently and to manage other daily life activities autonomously (such as personal hygiene, writing, and eating). The UMRS score pre-PER treatment was 102; after 3 months under PER treatment it was 50. 

*Neurophysiological investigations:* The first EEG was performed at the age of 16 years, showing mild and a general slowing of the background activity with epileptiform anomalies in the left temporal derivations. A JL-Back AVG disclosed a positive–negative, biphasic spike at the central electrodes, somatotopically corresponding to the left forearm muscles (Figure 5). The potential was recorded approximately 30 ms before the rectified myoclonic EMG signal. Nerve conduction studies and EMG resulted as normal. However, the right median SEP evaluation revealed a giant cortical response (>20 µV) (Figure 3) [7]. In addition, a LLR was recorded from the right thenar muscles by stimulating the right median nerve.

*Ophthalmological Examinations:* A comprehensive ophthalmologic evaluation was performed at the age of 17 years, including BCVA measured with ETDRS chart at 2 meters, slit-lamp biomicroscopy examination, fundus oculi exam with retinography, and finally OCT with Heidelberg Engineering. She had the following BCVA: 65 letters in the right eye (20/25) and 55 letters (20/40) in the left eye. Slit-lamp examination showed in both eyes an incipient lens cortical opacity. There was no nystagmus. Examination of the ocular fundus showed a macular cherry-red spot bilaterally, with diffuse dystrophy of the retinal pigment epithelium in the middle periphery (Figure 4). SD-OCT showed, bilaterally, a thickening with increased reflectivity of GCL (Figure 4). 

*Brain Images*: Brain MRI performed at the age of 17 showed only mild atrophy of the superior part of the vermis (Figure 1).

*Additional Investigations:* The patient’s mental status was within normal limits. Detailed biochemical analyses and lactate values were normal; autoimmune screening was negative.

*Genetics:* Molecular analysis for Unverricht–Lundborg disease (cystatin B, *CSTB)* was normal. Next-generation sequencing (NGS) for epilepsy genes performed at the age of 17 years revealed two described heterozygous mutations into *NEU1* gene, i.e., a frameshift mutation c.1208delG (p.Ser403ThrfsTer85 rs1301852124) inherited from her mother and a missense mutation c.982G>A (p.Gly328Ser rs534846786) inherited from her father. 

## 3. Discussion

We have reported the clinical, neuroradiological, ophthalmological, and electrophysiological course of four unrelated patients affected by type 1 sialidosis, bearing compound heterozygosity of *NEU1*, and described the long-term follow-up of their disease (Table 1 and Table 2).

*Clinical Course*: The common clinical presentation of patients affected by type 1 sialidosis includes postural and action myoclonus, ataxia, macular cherry-red spot, visual defects, and seizures [4]. Myoclonus and ataxia are the most prominent symptoms, being present in a range of 92–100% and 78–87.8% of patients [9,10]. Seizures are reported in 73.7% of patients, although they can remain rare, and the macular cherry-red spot is seen in 51.2–60% of patients and might be missing at least at the beginning of the clinical presentation [3,9,10]. 

Our patients were followed up over a long period of time and have a disease course ranging from 6 to 30 years. The onset was characterized by visual problems, frequent falls, tremor, and TCSs. During the disease course, myoclonus became the most prominent symptom, confirming it as the key feature of the disease. Myoclonus, together with ataxia, led to a severe disability, resulting in loss of self-sufficiency within a few years from onset (4–10 years). Dysarthria was present and severe in three out of the four patients, sparing the youngest one, which has the shortest disease course. Other cerebellar signs were seen, such as dysmetria and hypotonia (two out of four). TCSs again were reported in the eldest three, sparing the youngest one. Two patients also suffered from myoclonic seizures, which could be triggered by sensitive stimulations, light, or emotional stress. These could be for a very long period of time, leading to a myoclonic status epilepticus (patient #1) or very frequent and easily provoked seizures (patient #2). In both cases, the awareness was preserved. Despite the long-lasting history, the polytherapy, and the severe myoclonus all preserved their intellectual abilities as commonly reported. Only one showed impairment in executive functioning. Indeed, only 22.2% of the patients with type 1 sialidosis showed cognitive dysfunctions [9].

*Treatment:* The medications used in this case series are commonly used to treat PME and include ASMs commonly used for TCSs (VPA, LEV, CLZ, ZNS, PB) and less common drugs used specifically against myoclonus (acetazolamide, PIR). 

Three out of four patients benefited from PER add-on, which proved to be effective and safe even at doses as low as 4 mg/day. Patient #1 showed reduced seizure number, and this allowed her to better perform the elementary daily activities. Patient #2 showed a significant amelioration of writing and eating, as well as complete seizure control, particularly with regards to the brief but very frequent myoclonic attacks triggered by light, eye closure, and emotional stress. Patient #3 regained the possibility to manage all essential daily life activities independently. The benefit was also proved by the significant improvement of the neurophysiological evaluation (EEG–EMG polygraph) and UMRS. PER has been reported as effective and well-tolerated in different PMEs including type 1 sialidosis [11,12]. Our case series confirm these data and suggest that PER is a good therapeutic option for these patients. 

Regarding other therapeutic options, gene therapy would be the preferred approach as it could provide a stable long-lasting therapeutic correction. However, this option is not available at the moment for type 1 sialidosis patients. A recent research study has demonstrated that some pharmacological and natural dietary components, such as romidepsin and betain, by acting as histone deacetylase inhibitors, induce a consistent increase of the residual activity of mutant NEU1 in patients’ primary fibroblast. These compounds have not been administered in patients under controlled trials yet. However, this observation may also explain the potential differences among patients with the same genetic mutations that may be attributed to lifestyle, diet, and environmental factors, aside from genetic background [13]. 

*Neurophysiological Investigations:* According to the current knowledge and guidelines about PME, during the long-term follow-up of these patients, different neurophysiological examinations (EEG, JL-Back AVG, LLR, PEV, and SEP) were performed to support the diagnosis and to evaluate the progression of the disease. Although these techniques are essential at a certain time (i.e., diagnosis, epileptic status, or evaluation of drug response), they might not always be available and patients can get worn out during the examinations. The UMRS can be used as a quantitative clinical test, being able to examine all the myoclonus aspects including myoclonus at rest, stimulus sensitivity, action myoclonus, and functional tests [14].

*Ophthalmological Evaluation*: Two patients mentioned visual problems at the onset, with these becoming very severe in one of the patients (patient #1). The pathognomonic macular cherry-red spot was seen in two patients. However, while we can rule this out for patient #3 as repeated ophthalmological evaluation confirmed its absence, patient #1 could not be properly assessed because of comorbidities, especially opsoclonus. The absence of the macular cherry-red spot might have postponed the genetic investigation, leading to a type 1 sialidosis diagnosis in these patients. When present, the macular cherry-red spot was also associated with thickening of the peripapillary retinal nerve fiber layer and increased reflectivity of GCL, as demonstrated by the OCT. These findings have been reported for other cases in the literature [15,16]. Lens opacity was detected in three patients, confirming the fact that it is also a common sign in this condition [16,17].

*Neuroimaging*: Brain MRI can be normal at onset. Diffuse brain atrophy is commonly observed in the advanced stage of type 1 sialidosis. However, atrophy can be severe, mostly affecting the cerebellum, cerebellar peduncles, cerebral hemispheres, and corpus callosum [3,18], and in some cases, it may progress rapidly [19]. Lu et al. previously studied the brain functional MRI of 11 patients affected by type 1 sialidosis, demonstrating a compromised posterior visual pathway, and concluding that the posterior part of the brain could be extensively involved, in addition to the eye abnormality [20].

All of our patients showed cerebellar atrophy, mostly affecting the vermis, which seemed to be progressively in line with the cerebellar clinical symptoms. Although we did not specifically investigate the time course of the atrophy’s worsening in each case, we noted a gradient consistent with the disease duration of our patients being worst in the eldest one (patient #1), and mildest in the youngest (patient #4). We also noted bilateral gliosis of the cerebellar folia. The visual pathway was damaged in the most severely affected patient (#1). In fact, in this patient, a chiasm thinning and bilateral peritrigonal and deep occipital white matter hyperintensities were detected on MRI. A cerebral PET of the same patient showed reduced metabolism over the occipital cortex, consistent with the clinical severely affected sight and confirming the important involvement of the visual pathway other than the cerebellum. 

*Genetics:* The genetic diagnosis was made after years from onset (23 years for patient #1, 5 years for patient #2, 13 years for patient #3, and 4 years for patient #4). All of our patients were compound heterozygous, despite two having consanguinity in their family history. They all had a combination of one mild and one severe variant each. Patient #1 and #4 shared both variants (c.982G>A and c.1208delG), although they were not related. The genetic details of patient #1 have been reported previously [8]. The variant c.1208delG is a frameshift mutation, leading to a 69 amino acid insertion at the C-term of the protein, with the mutant protein not having enzymatic activity and not being transported to the lysosome [1,21]. Thus, this variant is usually associated with the more severe form, namely, type 2 sialidosis. In these patients, the presence of the variant c.982G>A, whose mutant protein retained 40% of the enzymatic activity, probably mitigated the phenotype [1,9,21,22]. The variant c.982G>A was also shared with patient #2, who is from the same region as patient #1 and patient #4. The other variant of patient #2, namely, c.272T>G is a missense mutation reported in type 2 sialidosis patients [1,21]. Patient #3 is from Turkey and does not share variants with the other three that are all from Italy. The genetic details of this patient have been reported previously [8]. The first variant, c.625delG, is a deletion, causing a frameshift and consequently a premature stop codon. It is known to be pathogenic, and the mutant protein would have a complete loss of sialidase activity [1]. The other variant, namely, c.914G>A, which is located in the same codon, has not been previously reported in other patients. 

## 4. Conclusions

We reported the long-term follow-up and detailed clinical history of four patients affected by type 1 sialidosis, supporting it with neurophysiological data, as well as brain and ophthalmological images. This is a retrospective study and, as such, data and images with the same time laps could not be provided. The patients were followed up at different stages by different clinicians. However, we believe that this case series brings a comprehensive realist long-term overview of the clinical course of patients with type 1 sialidosis. The images comprehend the different neurophysiological, neuroimaging, and ophthalmological aspects that can be encountered when dealing with a type 1 sialidosis patient. Some new insights can be highlighted in this work: (a) we report a long-term follow-up, demonstrating a high degree of motor and speech disability for all patients with relative preservation of cognitive functions; (b) the treatment with PER, was found to be safe and effective against TCSs and myoclonus, even at low doses, leading to a significant benefit in the patient’s daily life activities; (c) new brain findings were obtained, including bilateral gliosis of the cerebellar folia and the occipital–parietal white matter; (d) a new variant (c.914G>A) of *NEU1* gene was found, which has not been previously described in other patients.

Owing to the rarity of the condition, only a limited number of patients affected by type 1 sialidosis could have been reported. The subtle presentation of the initial symptoms may expose these patients to considerable potential misdiagnosis. Indeed, although the onset can be at the beginning of the second decade of life, the diagnosis is only made after some years when the combination of ataxia, visual disturbances, and myoclonus prompt the suspicion of type 1 sialidosis. As for other PME, a precision treatment does not exist yet, and patients may be exposed to inadequate treatment or even contraindicated treatment options [6] until a correct diagnosis is made. Therefore, we believe that a comprehensive (neurophysiological, neuroradiological, and ophthalmological) evaluation is essential to help shorten the time lag to genetic diagnosis and subsequently assist these patients the most. This may also serve in the future to establish precision medicine treatments and to prevent the progression of this rare condition. 

## Figures and Tables

**Figure 1 brainsci-10-00506-f001:**
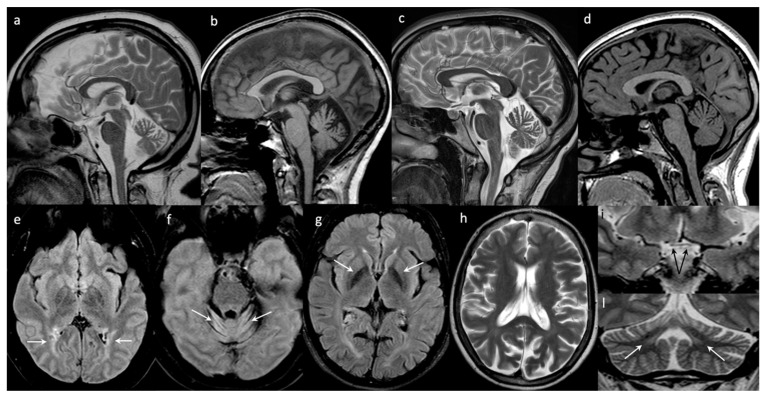
Brain magnetic resonance imaging (MRI) findings. Sagittal T2- ((**a**) patient #1; (**c**) patient #2) and T1-weighted ((**b**) patient #3, (**d**) patient #4) images of the four patients showing different degrees of cerebellar atrophy mostly involving the superior vermis and mainly dilating the primary fissure. Axial FLAIR images ((**e**) and (**f**) patient #1; (**g**) patient #3) demonstrate bilateral peritrigonal white matter hyperintensities (arrows on (**e**)), bilateral gliosis of cerebellar folia (arrows on (**f**)), and hypointense signal of the globi pallidi (arrows on (**g**)). Axial T2-weighted image shows diffuse brain atrophy, mostly involving parietal lobes ((**h**) patient #3). Coronal T2-weighted images of patient #1 demonstrate chiasm thinning (arrows on (**i**)) and, in the same patient, severe cerebellar volume loss associated with dentate nuclei hyperintensity (arrows on (**l**)).

**Figure 2 brainsci-10-00506-f002:**
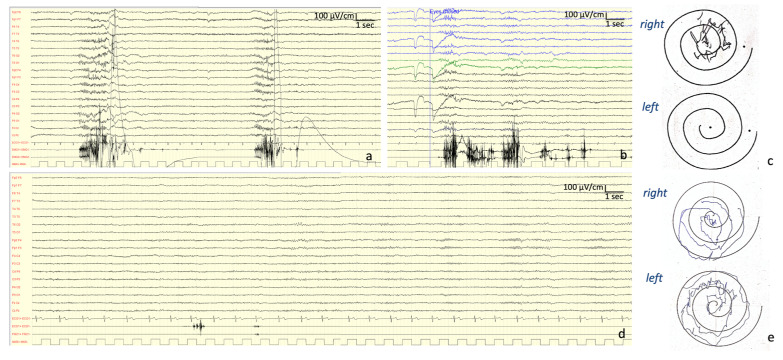
Elettroencephalogram and jerk-locked back averaging analysis (JL-Back AVG) of patient #2. (**a**,**b**) Polygraphy (electroencephalography (EEG)–electromyography (EMG)) performed at the age of 27 years, showing frequent discharges characterized by diffuse polyspikes on EEG channels accompanied by intense myoclonus (on EMG channels) recorded spontaneously (**a**), or provoked by eye closure (**b**). (**c**) Archimedes spiral (from unified myoclonus rating scale (UMRS), section 5; functional tests) performed at the same time. Note that the left hand could not perform the task. (**d**) Polygraphy (EEG–EMG) performed after 3 months from the introduction of perampanel (PER) showing the dramatic disappearance of the discharges, which was also accompanied by improvement of the Archimedes spiral (**e**).

**Figure 3 brainsci-10-00506-f003:**
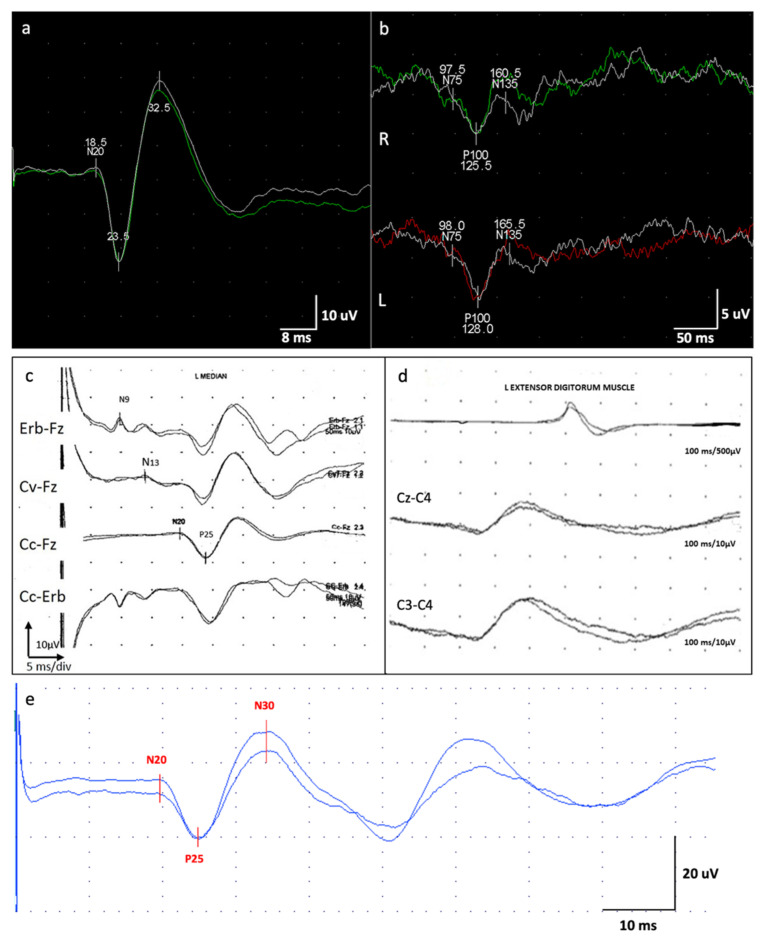
Neurophysiological findings: somatosensory evoked potential (SEP) and visual evoked potential (VEP). (**a**) Right median nerve SEP, patient #2. The cortical response recorded from somatosensory cortex C3’-C4’ showed extremely enhanced N20-P25 and P25-N30 amplitude (25 μV and 50 μV), configuring the typical wave of a giant SEP [7]. (**b**) A 30 min checkerboard reversal pattern VEPs of patient #2. The cortical response recorded from Oz-Fpz showed delayed latency of VEP components (N75, P100, N145) and reduced amplitude N75-P100 (R: 5 μV; L: 6 μV) for each eye stimulation. R: right stimulation; L: left stimulation. (**c**) Median SEP of patient #3 showing a giant cortical response (>20 µV); (**d**) JL-Back AVG analysis of patient #3 showing a biphasic right parietal wave approximately 20 ms before myoclonus potentials recorded from superficial electrodes located on the left forearm extensor muscles. (**e**) Right median nerve SEP of patient #4. The cortical response recorded from somatosensory cortex C3’-C4’ showed extremely enhanced N20-P25 and P25-N30 amplitude (18 μV and 50 μV, respectively), configuring the typical wave of a giant SEP [7].

**Figure 4 brainsci-10-00506-f004:**
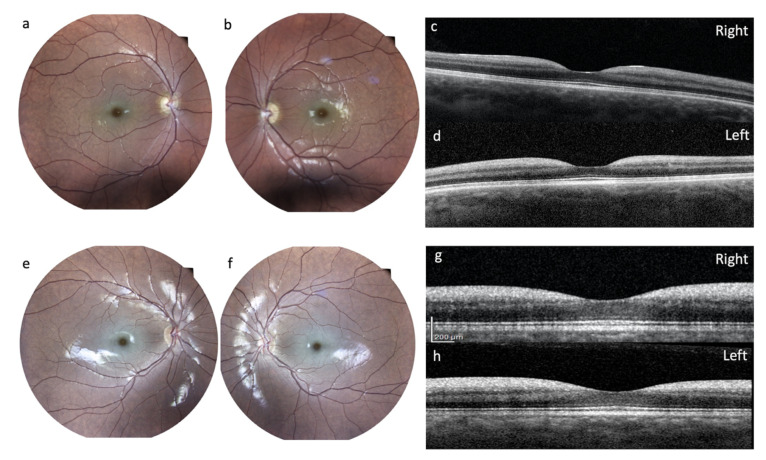
Ophthalmology findings. Ocular fundus of patient #2 (**a**,**b**) showed a macular cherry-red spot bilaterally, with diffuse dystrophy of the retinal pigment epithelium in the middle periphery. In the right eye (**a**) the optic disc was normal, while in the left eye (**b**), on the nasal portion of the optic disc, a vitreous thickening zone was evident. SD-OCT (**c**,**d**) showed a thickening with increased reflectivity of ganglion cell layer (GCL) in both eyes. Ocular fundus of patient #4 (**a**,**b**) showed a macular cherry-red spot bilaterally, with diffuse dystrophy of the retinal pigment epithelium in the middle periphery. SD-OCT (**g**,**h**) showed, bilaterally, a thickening with increased reflectivity of ganglion cell layer (GCL).

**Figure 5 brainsci-10-00506-f005:**
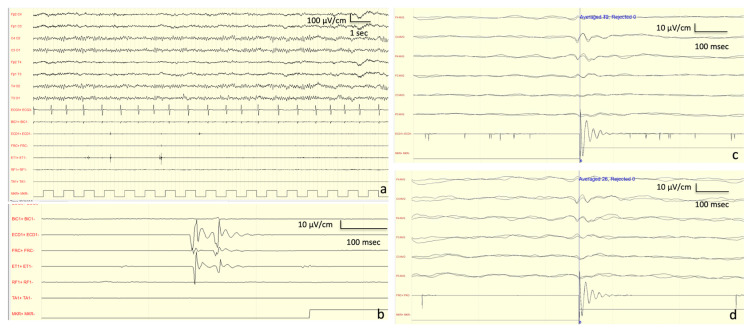
Elettroencephalogram and JL-Back AVG of patient #4. *(***a**) Polygraphy performed at the age of 16, showing a normal background EEG activity and spontaneous brief myoclonic jerks recorded over the EMG channels (BIC1: left bicep; ECD1: left extensor communis digitorum; FRC: left flexor carpi radialis; ET1: left first interosseous; RF1: left rectus femoris; TA1: left tibialis anterioris). (**b**) Same polygraphy showing a cranio-caudal progression of the jerks (ECD1 > FRC > ET1). (**c**,**d**) The results of the JL-Back AVG analysis performed by using the back averaging tool of the Micromed recorder (SystemPlus software; Micromed, Mogliano Veneto, IT). Analogical EMG triggers were manually selected on ECD1 (**c**) and on FRC (**d**). The images show the superimposition of two runs from two consecutive epochs (10 minutes each) of the same recording ((c) first run, 49 triggers; second run, 72 triggers; (d) first run, 26 triggers; second run, 28 triggers). The analysis disclosed a positive–negative, biphasic spike maximally observed at the central electrodes (C4-P4) and somatotopically corresponding to both the examined left forearm muscles.

**Table 1 brainsci-10-00506-t001:** Clinical, laboratory, and genetic findings.

Patient ID/Sex/Age (years)	Age of Onset; Age at Last Observation (years)	Symptoms at Onset	Neurological Signs Other than Cortical Myoclonus	Non-Neurological Features	ASMs at Last Follow-Up (dose/day)	Urinary Bound Sialic Acid ^(°)^	Alpha-Neuraminidase ^(°°)^	Mutations (Protein Change)	Genetic Details Reported
Patient #1 F/43	13; 43 (disease duration: 30 years)	Oscillopsia; frequent falls; GTCS	Severe ataxia; dysarthria; dysmetria; hypotonia; pyramidal signs; GTCS	Joint contractures; obstructive apnea syndrome; hypothyroidism; kidney and biliary stones	VPA 600 mg; LEV 2 gr; PB 100; CLZ 4 mg; PER 8 mg	93 nM/creatine M	0.55 nM/h/mg	**c.982G>A** (p.Gly328Ser) rs534846786 **c.1208delG** (p.Ser403ThrfsTer85) rs1301852124	[8]
Patient #2 F/26	13; 28 (disease duration: 15 years)	Frequent falls; tremor; hemeralopia	Ataxia; severe dysarthria; nystagmus; dysmetria; hypotnia; dysdiadochokinesia; GTCS	Cleft palate	PER 4 mg; LEV 1500 mg; acetazolamide 250 mg; CLZ 3 mg	60 nM/creatine M	0 nM/h/mg	**c.272T>G** (p.Leu91Arg) rs104893972 **c.982G>A§** (p.Gly328Ser) rs534846786	
Patient #3 F/42	17; 41 (disease duration: 24 years)	Tremor; myoclonic jerks; imbalanc; GTCS	Severe ataxia; dysarthria; dysmetria; tremor; executive functioning impairment; GTCS	none	ZNS 200 mg; LEV 3 gr; Primidon 500 mg; CLZ 2 mg	NA	NA	**c.914G>A** (p.Arg305His) rs774362886 **c.625delG** (p.Glu209SerfsTer94) novel	[8]
Patient #4 F/18	12;18 (disease duration: 6 years)	Postural tremor	Ataxia	none	LEV 1 gr; CLZ 1 mg; acetazolamide 250 mg; PER 4 mg	NA	NA	**c.982G>A§** (p.Gly328Ser) rs534846786 **c.1208delG** (p.Ser403ThrfsTer85) rs1301852124	

° normal values: <60 nM/creatine M; °° normal values: 100-390 nM/H/mg.

**Table 2 brainsci-10-00506-t002:** Ophthalmological, neurophysiological, and brain MRI findings.

Patient ID/Sex/Age (years)	Ophthalmological Findings	EEG Findings at Onset/Last Follow-Up	Other Neurophysiological Findings	Brain MRI
Patient #1 F/43	Opsoclonus, bilateral cataracts absent (?), macular cherry-red spot, glaucoma.	Normal/slow background, temporal spikes and polyspike-wave discharges, IPS.	Enlarged SEPs, Enhanced LLR JL-Back AVG + impaired VEP	Hyperostosis, peritrigonal white matter hyperintensities, severe cerebellar atrophy, dentate nuclei hyperintensity, vermian atrophy, bilateral gliosis of the cerebellar folia, thinning of the optic nerves and chiasm.
Patient #2 F/26	Diffuse opacity of the corneal endothelium; lens cortical opacities. Bilateral macular cherry-red spot. Diffuse dystrophy of the retinal pigment epithelium. Vitreous thickening. Thickening with increased reflectivity of ganglion cell layer in both eyes.	Normal background activity with interictal generalized epileptiform discharges/normal background activity, beta rhythms, diffuse polispikes (triggered by IPS and eye closure).	Enlarged SEPs, Enhanced LLR JL-Back AVG + impaired VEP and MEP	Cerebellar and ponto-mesencephalic atrophy.
Patient #3 F/42	Absent macular cherry-red spot (on repetitive examination).	Normal/slow background of theta range (7Hz), IPS.	Enlarged SEPs, enhanced LLR Back AVG + impaired VEP	Cerebral (parietal lobe) and cerebellar atrophy; hypointense signal of globi pallidi.
Patient #4 F/18	Lens cortical opacities. Bilateral macular cherry-red spot bilaterally; diffuse dystrophy of the retinal pigment epithelium. Bilateral thickening with increased reflectivity of ganglion cell layer.	NA/mild, diffuse slowing of the background activity, epileptiform anomalies over left temporal region.	Enlarged SEPs, enhanced LLR BACK AVG + impaired VEP	Mild cerebellar atrophy.

Abbreviations; ASMs: anti-seizure medications; CLZ: clonazepam; GTCS: generalized tonic–clonic seizures; IPS: intermittent photic stimulation; LEV: levetiracetam; NA: not available; LLR: long-loop reflex; PB: phenobarbital; PER: perampanel; PIR: piracetam;; SEPs: somatosensory evoked potentials; VEP: visual evoked potential; VPA: valproic acid; ZNS: zonisamide.

## Data Availability

All data generated or analyzed during this study are included in this published article (and its Appendix A).

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
