# Peer review of "Diagnosis and Management of Type 1 Sialidosis: Clinical Insights from Long-Term Care of Four Unrelated Patients"

_brainsci, 2020, doi:10.3390/brainsci10080506_

Round 1

Reviewer 1 Report

  1. Double check that when a name of name that will be subsequently abbreviated that both name/abbreviation are listed together the first time you use it. Example on page 3 line 101: Perampanel (PER)
  2. Page 2 line 26: Patient #1: this 43-year old Italian woman... Consider: A 43-year old Italian female...
  3. Page 2 Line 92: All that time, TCSs were rare... This sentence needs clarification.
  4. Page 9 Line 265: A 41-year old Turkish female (instead of woman) born to consanguineous parents (instead was born to related parents). Can you tell the degree of consanguinity?
  5. Page 9 Line 266: Consider At 17 years of age...Instead of "At 17 years...."
  6. Page 9 Line 271: "T" the age of 20 years....?
  7. Page 9 Line 278: "At last, follow up....Consider not started sentence with "at last..."
  8. Page 9 Line 281: CLN is not listed in the List of abbreviations
  9. Page 9 Line 294: Brain images: at onset...at age of onset of symptoms (17 years)?
  10. Page 9 Line 301:Consider disease or disease causing genes, although I consider using both ideal: Laforin (EPM2A), Malin (NHLRC1), CSTN for cystatin B (results in Unverricht-Lundborg disase
  11. Page 10 Line 309: Consider: An 18 year old Italian female instead of "This 18 year old Italian patient..."
  12. Page 13 Line 425: "The genetic diagnosis  was only made after few years..." Consider : The genetic diagnosis was made after years...

Reviewer 2 Report

Coppola et al. present the clinical, imaging, electrophysiological and molecular finding of four unrelated cases of Sialidosis type-1. 

The authors present a well rounded description of the most relevant clinical features including action myoclonus, ataxia, and seizures along the span of the disease over the years.  While, most of the reported findings are known, the fact that a rich phenotypic description across many domains has been done add value to this communication. 

The authors make an interesting claim regarding the therapeutic benefit of Perampanel on the therapeutic management of symptoms related to Sialidosis Type-1.  This is an important observation and one that perhaps needs to be treated in a more detail fashion in the manuscript.  While I realize this is an observational report on the effects of this medicine in three cases, it will be useful to expand, or even dedicate table or a portion of the discussion to the effects of Perampanel

a) benefits on activity of daily living,

b) seizure/myoclonus reduction (in a quantitative/semi-quantitative way),

c) side effects,

d) capacity to reduce/modify other medications, etc.

e) Duration of treatment observation and need for dose escalation

and perhaps even speculate on why this anticonvulsant has such benefits in the Sialidosis patient population.

Figure 2 is large and complicated.  Without magnification, it is difficult to view relevant features and could be improved.  For example, the traces from "jerk locked averaging" lack superimposition of  at least two runs to ensure reproducibility of waveforms and lack a corresponding polarity, voltage and time scales.  Some technical description of the methodology and instrumentation for electrophysiological studies.

Reviewer 3 Report

This manuscript gives a detailed account of the main clinical aspects and long-term care of 4 patients affected by the attenuated, type I form of sialidosis, a lysosomal storage disease of glycoprotein metabolism that is considered ultrarare and for which no target therapy is currently available.

This form of the disease is particularly problematic to diagnose and treat because patients are mostly asymptomatic early in life and later develop symptoms, such as myoclonus, that are often mistaken for those present in other neurological disorders. The four patients described here are no exception in this respect. They were all diagnosed years after the occurrence of the first symptoms and all via NGS or whole exome sequencing.

This add to the importance of this report that is both necessary and informative for the clinical and scientific community alike, given the limited number of patients and the lack of comparative assessment of the severity of the clinical manifestations and of treatment regimens especially in reference to medications used to control the myoclonus.

Particularly interesting is the spectrum of genetic mutations identified in the four patients. Of the three Italian patients, two of them are compound heterozygotes for the same NEU1 mutations and the third share with the former two    one allelic mutation. The long-term follow-up (6-30 years) and comparative clinical analyses of these three patients should inform on the course of specific clinical phenotypes in this form of sialidosis and on the potential differences in severity among patients with the same genetic mutations, differences that may be attributed to life style, diet, and other environmental factors, beside genetic background.

A recent publication (Mosca et al., J. Clin. Med. 2020) addresses these aspects that might be important to take into account during standard of care for sialidosis type I.  In this respect the authors might consider including this as discussion point. Lastly, for the sake of clarity and avoid confusion, any previous publications reporting the genetic studies of these four patients should be better incorporated in a separate column in Table 1.
